# Lipidomic Approaches in Common and Rare Cerebrovascular Diseases: The Discovery of Unconventional Lipids as Novel Biomarkers

**DOI:** 10.3390/ijms241612744

**Published:** 2023-08-13

**Authors:** Antonella Potenza, Gemma Gorla, Tatiana Carrozzini, Anna Bersano, Laura Gatti, Giuliana Pollaci

**Affiliations:** 1Laboratory of Neurobiology and UCV, Neurology IX Unit, Fondazione IRCCS Istituto Neurologico Carlo Besta, 20133 Milan, Italy; antonella.potenza@istituto-besta.it (A.P.); gemma.gorla@istituto-besta.it (G.G.); tatiana.carrozzini@istituto-besta.it (T.C.); giuliana.pollaci@istituto-besta.it (G.P.); 2Cerebrovascular Unit, Fondazione IRCCS Istituto Neurologico Carlo Besta, 20133 Milan, Italy; anna.bersano@istituto-besta.it; 3Department of Pharmacological and Biomolecular Sciences, Università di Milano, 20122 Milan, Italy

**Keywords:** biomarkers, lipids, mass spectrometry, cerebrovascular diseases, targeted lipidomic, untargeted lipidomic

## Abstract

Stroke remains a major cause of death and disability worldwide. Identifying new circulating biomarkers able to distinguish and monitor common and rare cerebrovascular diseases that lead to stroke is of great importance. Biomarkers provide complementary information that may improve diagnosis, prognosis and prediction of progression as well. Furthermore, biomarkers can contribute to filling the gap in knowledge concerning the underlying disease mechanisms by pointing out novel potential therapeutic targets for personalized medicine. If many “conventional” lipid biomarkers are already known to exert a relevant role in cerebrovascular diseases, the aim of our study is to review novel “unconventional” lipid biomarkers that have been recently identified in common and rare cerebrovascular disorders using novel, cutting-edge lipidomic approaches.

## 1. Introduction

### 1.1. Cerebrovascular Disorders

The brain is one of the most vascularized organs because of its unique need for oxygen and nutrients. Cerebrovascular diseases (CVD) consist of a heterogeneous group of conditions that share abnormalities in blood flow and/or brain vasculature. Circulation disturbances can result from narrowing of the blood vessels (stenosis), clot formation (thrombosis), arterial blockage (embolism) or vascular rupture (bleeding); all these conditions determine a lack of appropriate cerebral blood flow to thought brain tissue, leading to ischemic or hemorrhagic stroke or transient ischemic attacks (TIA). Cerebrovascular malformations, whether genetically determined or not, could lead to an inappropriate growth of vessels, which may become dysfunctional, degenerate or regress, as in neurodegenerative diseases [1]. The main manifestation of cerebrovascular disorders is, indeed, stroke. Stroke represents a highly prevalent and often devastating disease that primarily affects the elderly. Every year in Europe, about 1.1 million strokes occur, causing 440,000 deaths [2]. The Global Burden of Disease Study in 2019 reported 12.2 million incident cases of strokes, 101 million prevalent strokes and 6.55 million deaths from stroke worldwide. Globally, stroke was confirmed to be the second-leading cause of death (11.6% of total deaths) after ischemic heart disease and the third-leading cause of death and disability in 2019. Notably, stroke events are expected to increase dramatically due to the general ageing of the population [3,4].

Stroke manifests as two different subtypes, ischemic and hemorrhagic, both resulting in poor brain blood supply. Ischemic stroke (IS) is the common consequence of any occlusion of a blood vessel, which can be due to different factors: thrombotic or embolic occlusions or arterial stenosis determined by protein deposition and/or cellular proliferation (atherosclerosis above all). On the other hand, hemorrhagic stroke (HS) is characterized by the disruption of vessel walls, resulting in bleeding into the brain parenchyma (intracerebral hemorrhage, ICH) or in the subarachnoid space (subarachnoid hemorrhage, SAH).

Ethnicities are known to affect stroke etiologies. For example, Asian patients have a higher incidence of intracranial atherosclerosis disease, while in Western countries, extracranial carotid arteries plaques are predominant [5,6].

Since patients affected by IS or HS often exhibit similar clinical features, current diagnostic criteria rely on imaging techniques and the presence of risk factors. A proper clinical assessment is based on a thorough knowledge of anatomo-functional features of cerebral circulation.

At a microvascular level, cerebral perfusion after stroke greatly depends on the status of the collateral, particularly leptomeningeal circulation, which significantly varies between individuals and can be highly affected by vascular risk factors and diseases. At the macrovascular level, due to anatomical and disease-related factors, only 50% of the general population presents with an intact circle of Willis [7]. Thus, a proper diagnosis requires high expertise supported by advanced imaging tools [8].

Evidence collected by a standardized international case–control study confirmed that about 90% of the population-attributable risk of stroke is associated with ten potentially modifiable risk factors: hypertension, smoking, diabetes mellitus, physical activity, diet, psychosocial factors, abdominal obesity, alcohol, cardiac causes, and apolipoproteins [9]. Indeed, healthy lifestyle choices (avoiding smoking, moderate consumption of alcohol, daily exercise and healthy diet) are able to reduce the risk of stroke by nearly 80% [10].

Increased levels of LDL cholesterol and triglyceride are straightly correlated to atherosclerosis processes, thus leading to the narrowing of arteries and ischemic strokes. Interestingly, the Cretan Mediterranean diet, including olive oil, whole grains, fruits, vegetables and legumes, by lowering cholesterol and saturated fat, may reduce stroke by 40% or more in high-risk patients [11]. Furthermore, some epidemiological studies have registered an association between a high-fat diet (HFD) and cognitive impairment. Indeed, HFD was reported to impact cognition through deleterious effects on blood vessels, enhancing neurodegenerative processes [12,13]. Additionally, patients with metabolic syndrome (MetS), which comprises several vascular risk factors and metabolic abnormalities such as centrally distributed obesity, atherogenic dyslipidemia, high blood pressure and hyperglycemia, are characterized by a 45% increased risk of stroke recurrence and a 25% increase in all-cause mortality [14,15]. Intriguingly, stroke is associated with disturbed polyunsaturated fatty acids metabolism of arachidonic acid (AA), which represents one of the main components of glycerol phospholipid degradation and plays a crucial role in oxidative damage after stroke since free AA increases the production of free radicals through the arachidonic acid cascade reaction. Currently, a Mendelian Randomization (MR) study using genetic tools to predict plasma phospholipid AA supported the destructive role of AA on ischemic stroke, whereas suggested positive protective effects of α-linolenic acid (ALA) [16].

About 30% of stroke events remain of undetermined origin (“cryptogenic strokes”) [17]. Nevertheless, several rare heritable and non-heritable conditions often remain misdiagnosed, and their prevalence is usually higher in young patients [18]. The diverse etiologies of CVD subtypes would determine specific therapeutic treatments to prevent patients from recurrence of events. Therefore, it would be of great importance to identify new circulating biomarkers able to diagnose and classify CVD subtypes.

Many conventional lipid biomarkers (total cholesterol, triglycerides, low-density lipoprotein cholesterol LDL and high-density lipoprotein cholesterol (HDL)) have been widely discussed and have already been identified as CVD risk factors [19,20,21]. For example, hemodynamic features such as high blood pressure and low wall shear stress can increase the filtration rate of low-density lipoprotein (LDL) in vascular endothelium, accelerating the growth of atherosclerotic plaques and increasing the risk of in-stent restenosis [22]. However, although the cerebrovascular risk related to lipid disorder is well known, the specific lipid profile, e.g., the lipid molecules and their role in the process, is still unclear. Consequently, the clinical application of this knowledge is limited so far [23].

Recently, increasing evidence showed that unconventional lipid molecular species, such as phosphatidylcholine (PC), phosphatidylethanolamine (PE), phosphatidylinositol (PI), phosphatidylserine (PS) and sphingomyelin (SM), play essential roles in the development of IS [21]. Thus, the aim of the present study is to provide a comprehensive overview of non-conventional lipid biomarkers in common and uncommon CVD using novel, cutting-edge lipidomic approaches.

### 1.2. Lipid Biomarkers

The term “biomarker” refers to a wide range of biological and medical signs that can be measured accurately and reproducibly. Despite its enormous value, there is great confusion about its definition and its use in clinical research and practice [24]. The International Programme on Chemical Safety, led by the World Health Organization and in coordination with the United Nations and the International Labour Organization, expanded the concept of biomarker by defining it as “any substance, structure or process and also products that can be measured in the body and can influence or predict the incidence and outcome of the disease”. An even broader definition includes the effects of environmental processes, interventions and exposures, including blood pressure, imaging markers and specific analytes found in biofluids and tissues. In clinical practice, medical signs play a key role, and biomarkers can thus be considered the most objective and quantifiable medical signs that can be measured reproducibly through modern scientific laboratory approaches [25]. There are several subtypes of biomarkers, which are identified according to their alleged applications:○Diagnostic biomarker: it detects or confirms the presence of a disease or condition of interest;○Monitoring biomarker: it can be measured several times to assess the status of a disease, to monitor its natural progression or to detect an effect of a medical product or an environmental agent;○Predictive biomarker: it can predict response or future probable adverse events;○Prognostic/outcomes biomarker: it is used to identify the likelihood of a clinical event or disease recurrence or to estimate the progression of a disorder.

Given a biomarker that can be measured with sufficient precision and reliability with a delineated context of use, its use in clinical practice requires complex processes of qualification and validation. One crucial goal is to define a validation methodology that assures that the biomarker can be measured reliably, precisely and repeatably at a low cost. All too often, assays are not validated, engendering misleading assumptions about the biomarker’s value itself [24].

Circulating biomarkers are key factors that allow the identification of underlying disease mechanisms, often revealing a biological condition even before observable signs or symptoms appear. Early biomarkers represent an effective clinical option for timely diagnosis and for the selection of the best treatment option for each patient [26]. In this respect, new cutting-edge technologies with greater specificity and sensitivity represent a significant advance [27]. The advent of “omic” technologies allowed the identification of alterations in DNA, mRNA, microRNAs, proteins and metabolites from a single sample consisting of several biological matrices (e.g., peripheral blood, cerebrospinal fluid CSF, tissue biopsies). Carrying out comprehensive and representative profiles using these innovative strategies will help the development of ground-breaking personalized medicine approaches [27].

Lipids exert essential roles in biological systems, specifically membrane composition, energy storage and signalling. Lipidomic is the most powerful analytical tool to study lipids in biological specimens and their biochemical involvement in human diseases. Due to its sensitivity and selectivity, mass spectrometry (MS) represents the method of choice for qualitative and quantitative lipidomic analysis [28]. Indeed, MS enables the identification and quantification of hundreds of molecular lipid species in a short period of time, covering a wide range of lipid classes.

To date, enormous strides forward have been made with lipidomics, particularly in clinical medicine, by increasing the collective knowledge of the role of lipids in several diseases. Lipids have extensive information-bearing functions in the CNS as both ligands and substrates for proteins. The brain is indeed rich in lipids but has a poor capacity to synthetize them, so lipids must be supplied from peripheral blood circulation and cross the barriers within the CNS by using different strategies. Lipids cross the blood–brain barrier through passive diffusion and specific and non-specific transporters (transcytosis, involving LDL receptors and through a transport protein). All these mechanisms are finely regulated to maintain brain homeostasis. A perturbation due to any cerebrovascular disease might change the permeability to lipids taking to a modification of the brain lipid homeostasis. As an example, defective cholesterol and fatty acid homeostasis in the brain have been associated with age-related diseases, especially neurodegenerative ones [29]. Since lipid metabolism is often compromised in different brain diseases, the identification of putative lipid biomarkers seems crucial to better characterize the various disorders [30]. Many clinico-pathological fields have significantly benefited from lipidomic studies (e.g., cardiovascular disease, cancer and neurodegenerative diseases). Table 1 details the main techniques that are currently used to study human and animal lipidomic profiles. This includes associated lipid profiles that have been identified: some of them are currently exploited in the clinical setting, while others might provide suitable biomarkers or potential druggable targets for future investigation [31,32].

Besides biochemical measurement in biofluids, the lipid component is also important in atherosclerotic plaque formation; accordingly, recent advances in imaging techniques, including high-resolution MRI, enabled the direct observation of lipid plaques. More than one review [33,34] determined the predictive value of carotid intraplaque hemorrhage (IPH) on cerebrovascular events by MRI, associating the presence of IPH with a higher risk (approximately 6-fold higher) for cerebrovascular events.

Here, we will focus our attention on the search for non-conventional lipid biomarkers in common and rare CVD by using omic techniques/approaches.

**Table 1 ijms-24-12744-t001:** The main mass spectrometry techniques used to study lipidomic profiles in human and animal models.

Lipidomic Studies Techniques		Disadvantages	Humans	Animals
Liquid Chromatography (LC-MS/MS)	Analytical technique that involves physical separation of target compounds (or analytes) followed by their mass-based detection.	Expensive and time-consuming approach.	[35,36,37]	[38,39,40]
Shotgun Lipidomics (DI-MS)	Different instrumental platforms operated in direct infusion. Mostly relying on electrospray ionization (ESI).	Ion suppression effects.	[41,42]	[42,43]
Gas Chromatography (GC/MS)	High-resolving power and ability of the MS to provide precise data for identification and quantification of the separated substances.	Inaccurate quantification and identification of non-volatile compounds.	[44,45,46]	[47,48,49]
Nuclear Magnetic Resonance (NMR), Matrix-Assisted Laser Deposition/Ionization (MALDI)	A “soft” ionization technique, since it causes minimal or no fragmentation, which allows the identification of analytes molecular ions, even in complex mixtures of biopolymers.	Low Sensitivity. Lack of automatization and missing embedding into high-throughput lipidomic workflows.	[50,51]	[52,53]

## 2. Methods

To conduct a comprehensive review, a literature search was carried out using electronic databases, including PubMed, Scopus and Web of Science, to identify all the relevant studies published between 2000 and 2023. The search was devised using a combination of terms, including “biomarkers; lipids; mass spectrometry; cerebrovascular diseases; targeted lipidomic; untargeted lipidomic; cerebrovascular disorders; lipidomics; sphingolipid; lipid metabolism; stroke; cerebral small vessel disease; cerebral amyloid angiopathy, large artery atherosclerosis, Fabry Disease, Cerebral Autosomal Dominant Arteriopathy with Subcortical Infarcts and Leukoencephalopathy; Moyamoya Angiopathy”. The search was restricted to articles published in English. The final reference list was generated based on the relevance of each article to the topics discussed in this review (Figure 1).

## 3. Lipid Biomarkers in Common Cerebrovascular Diseases

Many recent studies have highlighted that the discovery of “unconventional” circulating lipid biomarkers in common cerebrovascular diseases is a basic condition for identifying novel diagnostic and therapeutic targets. The present section will focus on some common CVD: cerebral amyloid angiopathy (CAA), cerebral small vessel disease (CSVD) and large artery atherosclerosis (LAA).

### 3.1. Cerebral Amyloid Angiopathy

CAA is a form of cerebral small vessel disease and the second most common cause of cerebral hemorrhages after hypertension. It is determined by the progressive deposition of amyloid substance in the wall of cortical and small-to-medium-sized leptomeningeal arteries and, less frequently, in the capillaries and cerebral veins. Clinical presentation includes spontaneous major (HS due to intracerebral hemorrhage) or microscopic (microbleeds) hemorrhagic events, transient neurological focal events and cognitive impairment [54]. CAA diagnostic criteria and management are still under discussion due to the partial overlap of CAA clinical manifestation with Alzheimer’s disease (AD).

Using a well-characterized mouse model of sporadic CAA, the presence of different lipids, such as PCs and lysophosphatidylcholines (LPCs) in plasma samples was investigated. As a result, the levels of eight diacyl PCs (PCaaC30:2, PCaaC34:1, PCaaC36:1, PCaaC38:4, PCaaC38:5, PCaaC38:6, PCaaC40:4, PCaaC40:6), two acyl-alkyl PCs (PCaeC38:0, PCaeC40:4) and five LPCs (lysoPC C16:0, lysoPC C16:1, lysoPC C18:0, lysoPC C18:1, lysoPC C20:4) were found significantly increased. This is the first lipidomic study performed on a CAA mouse model and the first time when a lipid disbalance has been clearly identified in this pathological condition. Despite some peculiarities of the mouse model that can prevent human translatability, the fifteen plasma lipids indicated above may be a valuable starting point to deeply characterize CAA, providing the basis for further studies in humans [55].

### 3.2. Large Arteries Atherosclerosis

LAA is responsible for approximately 15% of all IS and is one of the main causes of transient ischemic attacks (TIA). It is mainly due to atherosclerotic stenosis or occlusion of the major intracranial arteries, and it can be classified into four distinct clinical scenarios: asymptomatic and symptomatic extracranial carotid stenosis, intracranial atherosclerotic disease, and extracranial vertebral artery atherosclerotic disease. Multiple factors contribute to the pathogenesis of atherosclerosis, such as endothelial dysfunction, inflammatory and immunologic factors, plaque rupture, as well as the common risk factors (e.g., hypertension, diabetes, dyslipidemia, smoking). Clinical manifestations of LAA differ based on lesion location [56]. Two studies focused on targeted lipidomic analysis of plasma samples from LAA patients aimed at identifying circulating plasma biomarkers typical of different stroke subtypes to provide an easier and better diagnosis. You et al. carried out a targeted lipidomic analysis of sphingolipid species in plasma samples [57]. The analyses showed that ceramides including Cer (d36:3), Cer (d34:2), Cer (d38:6), Cer (d36:4) and Cer (d16:0/18:1) were increased in LAA as compared to the control group, showing an increase of ceramides with a sensitivity above 80% and a specificity above 85%. Ceramides act as second messenger molecules and structural components of LDL, and they contribute to inflammatory pathways, oxidative stress and multiple pathological processes in atherosclerosis. Ceramides may exert a pathogenic effect on LAA, thus representing potential therapeutic targets, especially for those patients in whom the therapeutic effects of statins have failed.

Similarly, Wang and colleagues, by using targeted lipidomic, screened phospholipid content in plasma of LAA and cardioembolic (CE) stroke, the two most important subtypes accounting together for half of acute IS cases. They focused on plasma phospholipids (PLs) to assess their possible usefulness as diagnostic biomarkers. This study highlighted that the LAA group was characterized by higher peak areas related to four PL classes—PC; phosphatidylethanolamine, PE; sphingomyelins, SM and phosphatidylinositol, PI—when compared to a healthy control (HC) group, whereas by lower levels than the CE cohort. Using a supervised partial least squares-discriminant analysis (PLS-DA), the authors found a total of 14 crucial PL molecular species that might significantly contribute to the group differences. In particular, PC (18:0/18:2), PI (18:0/18:2) and PE (18:0/18:2) showed statistically significant differences between the IS group (LAA and CE patients) and the HC group but showed no difference among LAA and CE cohorts. Ten PL metabolites (SM(d18:1/18:1), PC (16:0/18:1), SM (d18:1/18:0), SM (d18:1/24:1), SM (d18:1/16:1), SM (d18:1/22:1), SM (d18:1/24:2), SM (d18:1/16:0), PC (16:0/18:2) and PC (16:0/22:6), were significantly different between all groups. They finally evaluated their diagnostic abilities through the ROC curve: SM (d18:1/18:1), SM (d18:1/18:0), SM (d18:1/24:1) and SM (d18:1/22:1) with AUC value at 0.903, 0.939, 0.936 and 0.909, respectively, were suggested to be exploited as qualified criteria in the differential diagnosis of CE and LAA [58].

### 3.3. Cerebral Small Vessel Disease

CSVD is determined by lesions of the small cerebral arteries and arterioles (diameter 40–200 µm), capillaries and venules supplying the white matter and deep structures of the gray matter [59]. CSVD is mostly sporadic, and accounts for approximately one-quarter of all IS, with a high prevalence in the Chinese rather than the Western population. Its occurrence is mainly associated with age, with no significant differences according to sex. Hypertension and diabetes mellitus are the most common risk factors, together with current and former smoking habits, obstructive sleep apnea and chronic kidney disease; from a microscopic point-of-view, lipohyalinosis and branch atheromatous disease represent the major histopathological disease correlates. The disease may remain asymptomatic for a long time, and it can only manifest through radiological examination [60]. In its acute form, it manifests with lacunar strokes or intracerebral hemorrhage [18]. The aforementioned study by You and colleagues analyzed lipid contents also in CSVD patients. They showed that sphingomyelins (SM) such as SM (d34:1) and several ceramides, including Cer (d34:2), Cer (d36:4), Cer (d16:0/18:1), Cer (d38:6), Cer (d36:3) and Cer (d32:0), were found increased in age-related CSVD when compared to control subjects, indicating that CSVD patients may be characterized by relevant changes of SM and Cer. In addition, when comparing the sphingolipid content between age-related CSVD and LAA, the authors demonstrated that the levels of SM (d34:1) and Cer (d36:4) were higher in the first group. Thus, SM (d34:1) and Cer (d36:4) may reasonably represent valuable biomarkers in distinguishing the two subtypes [57]. Because CSVD often dissimulates its onset, gradually progresses and seriously affects patients’ quality of life, these two sphingolipids may serve as putative biomarkers, providing clues for further investigation of CSVD pathogenesis.

Moreover, this study highlighted the crucial role of sphingolipids in possibly distinguishing pathogenic mechanisms and enhancing differential diagnosis between LAA and CSVD. Sphingolipids seemed to increase in LAA, while a relevant disturbance of SM metabolism has been observed in age-related CSVD.

## 4. Lipid Biomarkers in Rare CVD

Unconventional lipid biomarkers also emerged in rare CVD; here, we will focus on Fabry Disease (FD), Cerebral Autosomal Dominant Arteriopathy with Subcortical Infarcts and Leukoencephalopathy (CADASIL) and Moyamoya Angiopathy (MMA).

### 4.1. Fabry Disease

FD is a rare X-linked lysosomal storage disorder (LSD) which results in a decreased activity of lysosomal alpha-galactosidase-A (α-Gal A) enzyme causing the accumulation of glycosphingolipids, mostly globotriaosylceramide (Gb3), globotriaosylsphingosine (lyso-Gb3), as well as galabiosylceramide (Ga2) and their isoforms/analogs in the vascular endothelium, nerves, cardiomyocytes, renal glomerular podocytes and biological fluids [61]. The prevalence of FD ranges between 1/3100 and 1/117,000 across Europe due to highly heterogenous phenotypes potentially affecting its precise estimation [62]. Clinical presentation depends essentially on residual α-Gal A activity and patients’ sex. Two subtypes have been established: (i) the classic severe form, mostly affecting male patients; (ii) the asymptomatic or less severe subtype involving female subjects. Symptoms include episodic intermittent pain crisis (acroparestesia), vascular lesions (angiokeratomas), vascular tortoises and brown subepithelial linear deposits (cornea verticillata) of eyes, corneal opacities, hypohidrosis or anhidrosis, and proteinuria [63]. In adulthood, the disease progresses with cardiac, renal and cerebral complications. Of note, cerebral white matter lesions and occlusive cerebrovascular events might precisely occur due to Gb3 deposition.

Sphingolipidomic analyses in FD patients revealed valuable insights. Therefore, the study by You and colleagues was aimed at evaluating sphingolipids in the plasma of patients with FD to find out if there was any difference in sphingolipid content compared to age-related CSVD [64]. FD patients were characterized by higher trihexosylceramides (triHexCer) and SM (d34:1) levels compared to a control group, suggesting a putative SM metabolism abnormality. Additionally, plasmatic levels of Cer (d18:0/16:0) and several glycosphingolipids including triglycosylceramides (CerG3), such as CerG3 (d18:1/14:0), CerG3 (d18:2/16:0), CerG3 (d18:1/16:0+O), CerG3 (d18:1/16:0), CerG3 (d18:1/24:1), and CerG3GNAc1 (d36:2), and dihexosylceramides, such as CerG2 (d18:1/16:0+O) and CerG2GNAc1 (d32:1), were found enhanced in FD patients versus the control group, and CerG2GNAc1 (d41:4) decreased.

By comparing FD patients and age-related CSVD, this study highlighted that the first group is characterized by glycosphingolipid changes, while the most relevant alteration in the second one refers to Cer species. Notably, it is currently established that the vascular involvement of FD is mainly due to the deposition of glycosphingolipids in endothelial cells and vascular smooth muscle cells (VSMC), presumably because of the induction of oxidative stress [64]. Nevertheless, no significant differences in the severity or spatial distribution of lacunar infarction and white matter hyperintensities (WMH) in FD and age-related CSVD neuroimaging have been identified to date. Therefore, the detection of Cer and glycosphingolipids might provide a differential diagnosis between FD and age-related CSVD.

A recent lipidomics approach has been performed on renal tissue biopsy belonging to an FD patient to obtain a potential profile of lipid biomarkers, such as globotriaosylceramide (Gb3), globotriaosylsphingosine (lyso-Gb3), galabiosylceramide (Ga2) and their isoforms and analogues. As a result, the Gb3 and lyso-Gb3-related putative biomarkers were not detected in FD patients’ tissue. Conversely, the content of Ga2-related isoforms/analog was found to be enhanced in patients’ renal tissue biopsy when compared to two control samples. In particular, the ratio between the increase of Ga2[(d18:1) (C16:0)] and the corresponding ceramides decrease [Cer(d18:1) (C16:0)] was found to be 20 times higher in the patient’s sample versus the two controls.

This evidence confirmed that lipidomic analyses might contribute to the diagnostic evaluation when conventional anatomical and clinical pathological methods fail to provide a well-defined diagnosis [65].

### 4.2. Moyamoya Angiopathy

MMA is a rare and progressive CVD characterized by steno-occlusive lesions of the terminal part of the internal carotid arteries (ICAs), typically associated with a compensatory network of collateral vessels at the base of the brain [65]. Patients affected by MMA may undergo recurrent IS and/or HS, severe neurological deficits, physical disabilities and even death [66,67]. Many polymorphisms of the Ring Finger Protein 213 (RNF213) gene have been reported as susceptibility factors for MMA in the East Asian population, while no major susceptibly variants have been yet identified in Caucasian patients [68]. In the absence of a clear pathogenic disease picture, an impairment in etiopathology angiogenic and vasculogenic processes as potential. An untargeted lipidomic analysis revealed that the MMA lipid profile was homogeneous but deeply separated from that of healthy donor (HD) subjects [69]. Interestingly, a surprisingly lower content of lipids in plasma samples of MMA patients in comparison to HD was found. This depletion was mainly due to glycosphingolipids and phospholipids, which are plasma membrane-related components. The mostly deregulated lipids refer to the predominant human plasma long-chain GM3 molecular species such as ganglioside GM3 18:0, which, thanks to its role in the innate immune function of macrophages, promotes angiogenesis. Specifically, long-chain GM3 acts as an anti-inflammatory Toll-like receptor 4 (TLR4) modulator [70]. Sulfatide is reported to exert an anti-inflammatory role, likely by hindering the co-localization of TLR4 and lipid rafts [71]. Thus, the inflammatory features of MMA seem to be sustained by the decrease of both GM3 18:0 and SULF 16:0.

This study also highlighted an unexpected increase of cardiolipin [72], a peculiar mitochondrial lipid, in MMA plasma, thus supporting the mitochondrial abnormalities largely documented in MMA circulating endothelial cells [72,73]. Moreover, the authors underlined the depletion of one of the major Cer species implicated in defective angiogenesis, such as long-chain Cer 24:1. This evidence correlates with the increase of sphingosine-1-phosphate (S1P) and dihydrosphingosine-1-phosphate (DHS1P), the most abundant sphingoid bases found in plasma, presumably corresponding to the release of pro-angiogenic and cell growth-inducing lipid biomarkers from the newly produced MMA vessels. So far, many recent studies have revisited the relevance of MMA-related lipid metabolism. Since investigating the direct role of RNF213 in lipid droplet triglyceride accumulation [74], a recent study assessed RNF213 as an important modulator of lipotoxicity [75]. Dyslipidemia has been combined with symptomization of asymptomatic MMA patients in a work carried out by Hirano et al. [76]. Thus, a potential therapeutic strategy for MMA clinical treatment could be represented by targeting lipid metabolism. Indeed, searching for further correlation with clinical data may sustain a key role of lipids in MMA, potentially leading to the detection of reliable biomarkers and the identification of novel therapeutic targets.

### 4.3. CADASIL

CADASIL is a rare hereditary small vessel disease (SVD) caused by mutations in the NOTCH3 (Notch Homolog 3) gene situated on chromosome 19. It is currently defined as the most common genetic cause of stroke and dementia in adults. Pathogenetic mutations alter the number of cysteine residues in the extracellular domain of NOTCH3, a transmembrane receptor expressed by VSMC and pericytes, determining the extracellular accumulation of misfolded receptors in the small arteries and thus causing ischemic brain events. The main symptoms of CADASIL are migraine with aura, subcortical ischemic events, mood changes, apathy, cognitive impairment, progressive white matter degeneration and epilepsy [77]. The diagnosis is usually quite complex due to a relative lack of disease awareness in the clinical community and the heterogeneity of the disorder, even among family members affected by the same mutation. The diagnostic assessment generally occurs through a combination of NOTCH3 genetic testing; family history of migraines, strokes, and dementia; and MRI findings of diffuse white matter changes [78]. Recently, Sabogal-Guáqueta et al. carried out a study aimed at evaluating molecular changes in the lipidome signature of both sporadic Alzheimer’s disease (SAD) and CADASIL patients [79]. This study primarily sought to better understand the similarities between the alteration of lipid metabolism and pathogenesis of SAD and CADASIL, as well as the role of PL biomarkers in both brain disorders. Since PLs play a fundamental role in the brain metabolic network, their profiles were evaluated in the frontal cortex, white matter and CSF samples belonging to SAD and CADASIL patients and HC subjects. A comparable disbalance in the cerebral cortex district in both dementia groups (SAD and CADASIL) was found. Indeed, PC and PE species showed an increasing trend while phosphatidylserine (PS) decreased within the gray matter frontal cortex and CSF. Conversely, an inverse relation of lipidic changes (PC, PE reduction and PS increase) was detected in the white matter of both dementia groups. The authors obtained a significant model for distinguishing CADASIL and SAD patients from controls. This exploratory study represents the first experience of performing lipidomic analyses on CADASIL patients and provides a comprehensive overview of lipid profile modifications that might serve as putative biomarkers for the diagnosis of presymptomatic dementia with vascular involvement.

To offer a comprehensive overview of the literature about unconventional lipid biomarkers in common and uncommon CVD, we reported the main studies performed with omics techniques in Table 2.

## 5. Conclusions

Bioactive lipids interact in a complex metabolic network, playing key roles in biological systems. Since lipid metabolism might be strongly impaired and hindered when a neurological disorder occurs, the identification of putative lipid biomarkers is highly needed to empower diagnosis, prognosis and hopefully to propose tailored therapeutic options. So far, thanks to novel lipidomic approaches, a significant amount of valuable lipid markers have been identified in common as well as in rare CVD. This overview focused on the major unconventional lipid classes involved in cerebrovascular disorders, setting the ground to address future studies towards the most promising lipid classes/species. Thereby, unconventional lipids may extend the classic lipidome to identify diagnostic, prognostic and predictive biomarkers for CVD management. The lipidomic studies here reported have clarified the main roles of signaling, structural and metabolic lipids in the context of rare and common CVD. The omic profiling of such lipid molecules may represent an innovative approach that could fill some gaps in knowledge concerning these disorders.

Further investigations supported by innovative techniques as well as artificial intelligence and machine learning tools, are needed to broaden current knowledge and to translate the assessment of lipid biomarkers into clinical practice.

## Figures and Tables

**Figure 1 ijms-24-12744-f001:**
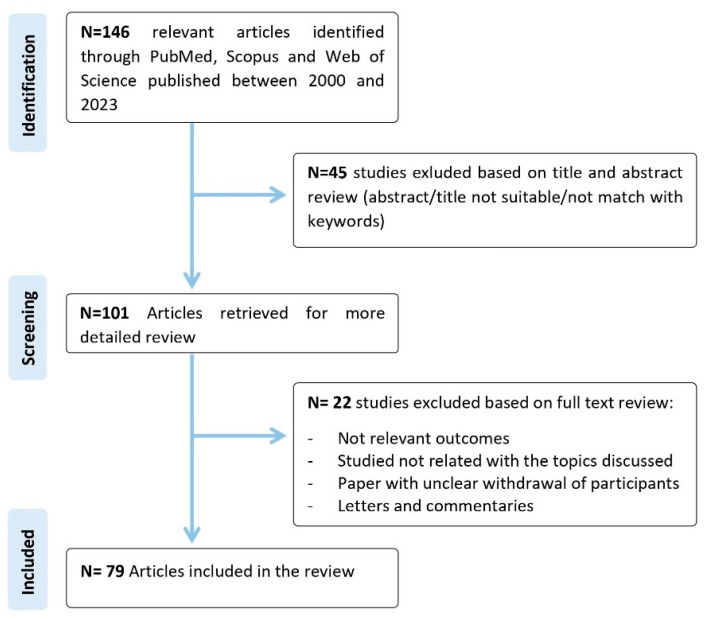
Flow chart of the literature screening.

**Table 2 ijms-24-12744-t002:** Main studies on the discovery of unconventional lipid biomarkers in common and uncommon cerebrovascular diseases through omic techniques, both in humans and in animal models. Cardioembolic, CE; ceramide, Cer; cerebral amyloid angiopathy, CAA; cerebral small vessel disease, CSVD; dihydrosphingosine-1-phosphate, DHS1P; galabiosylceramide, Ga2; globotraosylceramide, Gb3; globotriaosylsphingosine, lyso-Gb3; large artery atherosclerosis, LAA; lysophosphatidylcholines, LPCs; monosialodihexosylganglioside, GM3; moyamoya angiopathy, MMA; phosphatidylcholines, PCs; phosphatidylethanolamine, PE; phosphatidylinositol, PI; phosphatidylserine, PS; phospholipid, PL; sphingolipids, SLs; sphingomyelins, SM; phosphatidic acid, PA; sphingosine-1-phosphate, S1P; sporadic Alzheimer’s disease, SAD; subarachnoid hemorrhage, SAH; sulfatides, SULF; triglycosylceramides, CerG3; trihexosylceramides, triHexCer; lysophosphatidylethanolamine, LPE.

Author, Year, Ref	Disease	Tissue/Matrix	Number of Subjects	Techniques	Unconventional Lipids Alteration
[57]	LAA, CSVD and FD	Plasma samples	20 LAA patients; 20 patients with age-related CSVD; 10 FD patients; 14 controls	Targeted lipidomics: ultra-high-performance liquid chromatography quadruple-time-of-flight mass spectrometry/mass spectrometry	(i) Cer (d36:3), Cer (d34:2), Cer (d38:6), Cer (d36:4) and Cer (d16:0/18:1) were increased in LAA. (ii) SM (d34:1), Cer (d34:2), Cer (d36:4), Cer (d16:0/18:1), Cer (d38:6), Cer (d36:3) and Cer (d32:0) were increased in age-related CSVD compared to controls. (iii) Cer (d36:4) and SM (d34:1) were increased in age-related CSVD compared with LAA. (iv) Total trihexosyl ceramides were increased in Fabry group compared with control. SM (d34:1) was increased in FD patients.
[58]	LLA and CE	Plasma samples	58 LAA patients (mean age 67.4 ± 12.5), 19 CE patients (mean age 71.1 ± 12.9), 12 HC (mean age 61.9 ± 14.2)	Targeted lipidomic: triple-quadrupole mass spectrometer coupled with the HPLC system for mass spectrometric analysis	Seven molecular species of sphingomyelins (SM d18:1/18:1, d18:1/18:0, d18:1/24:1, d18:1/16:1, d18:1/22:1, d18:1/24:2 and d18:1/16:0), three molecular species of phosphatidylcholines (16:0/18:1, 16:0/18:2 and 16:0/22:6) showed significant differences in LAA, CE and healthy control (HC) groups.
[61]	Fabry disease	Renal tissue biopsy	1 patient, 2 control kidney specimens	Untargeted lipidomic: Ultra-high-pressure liquid chromatography high-resolution mass spectrometry (UHPLC-HRMS)	Increased level of Ga2[(d18:1) (C16:0)]; ceramide[(d18:1) (C16:0)] level decreased; Ga2[(d18:1) (C16:0)]/Ceramide[(d18:1) (C16:0)] ratio increased more than 20 times in the patient sample compared to the two control samples.
[69]	Moyamoya angiopathy	Plasma samples	40 MA adult Caucasian patients (mean age of 45.4 ± 12.4 years); 35 HD recruited as controls (mean age of 41.6 ± 11.9 years)	LC-MS/MS consisting of a Shimadzu UPLC coupled with a Triple TOF 6600 Sciex	MA significantly depleted in Cer 24:1, GM3 18:0, SULF 16:0, HexCer, LacCer, Gb3, LPC, LPE, PC, PE, PI, EtherPC, EtherPE and DAG.
[79]	CADASIL	Post-mortem tissue and CSF	5 subjects with SAD, 5subjects with CADASIL and 5 control subjects,	Automated ESI-MS/MS method was used, and data acquisition and analysis were carried out at the Kansas Lipidomics Research Center using an API 4000™ and Q-TRAP (4000QTrap) detection system	Significant increase in PC and reduction in LPC. Increased concentrations of PE and LPE and decreased concentrations of PI, PS, PA and PG.
[55]	CAA	Plasma samples of CAA mouse model	Sporadic CAA mice (n = 8) vs. control CAA mice (n= 8)	Targeted metabolomics: AbsoluteIDQ p150 Kit	Eight aaPCs (PCaaC30:2, PCaaC34:1, PCaaC36:1, PCaaC38:4, PCaaC38:5, PCaaC38:6, PCaaC40:4, PCaaC40:6), five LPCs (lysoPC C16:0, lysoPC C16:1, lysoPC C18:0, lysoPC C18:1, lysoPC C20:4) and two aePCs (PCaeC38:0, PCaeC40:4) were significantly elevated compared to the controls.

## Data Availability

Not applicable.

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
