# Peer review of "Lipidomic Approaches in Common and Rare Cerebrovascular Diseases: The Discovery of Unconventional Lipids as Novel Biomarkers"

_ijms, 2023, doi:10.3390/ijms241612744_

Round 1
Reviewer 1 Report (Previous Reviewer 1)
Dear Authors.
The manuscript you have submitted is a real breakthrough in lipidomics and its potential uses in research and the clinic. You performed an exhaustive background search for the review you submitted. You developed each section of the manuscript very well chosen and ordered, which made the reading of the document enjoyable and exciting. Just as a way of contributing to such excellent work. I want to make some suggestions:
1.- The tables used by you in the manuscript look very crowded in the printed version; please, separate the columns of the tables a little and decrease the font size so that the reading is clear.
2.- You must define the properties of a biomarker and its characteristics. You look for early biomarkers and look at studies that cite changes in lipid patterns or changes in quantity. These are excellent indices, but from there to being able to be used as a biomarker, a lot of work is involved in the development of biomarkers. Please discuss a little more the characteristics or properties of a biomarker so that the reader of your paper can come to the same conclusions you have made. I suggest a couple of reviews where you address the topic of biomarkers.
a. Califf RM. Biomarker definitions and their applications. Exp Biol Med (Maywood). 2018 Feb;243(3):213-221. doi: 10.1177/1535370217750088. PMID: 29405771; PMCID: PMC5813875.
b. Strimbu K, Tavel JA. What are biomarkers? Curr Opin HIV AIDS. 2010 Nov;5(6):463-6. doi: 10.1097/COH.0b013e32833ed177. PMID: 20978388; PMCID: PMC3078627.
Author Response
Dear Authors.
The manuscript you have submitted is a real breakthrough in lipidomics and its potential uses in research and the clinic. You performed an exhaustive background search for the review you submitted. You developed each section of the manuscript very well chosen and ordered, which made the reading of the document enjoyable and exciting. Just as a way of contributing to such excellent work. I want to make some suggestions:
We are grateful to the Reviewer for appreciating the innovative concept of lipid biomarkers in cerebrovascular disorders, and how it can really be a novel approach in biomedicine.
1.- The tables used by you in the manuscript look very crowded in the printed version; please, separate the columns of the tables a little and decrease the font size so that the reading is clear.
We thank the Reviewer for this suggestion. We modified the format of the tables, we decreased the text size and separated the columns for an easy reading.
2.- You must define the properties of a biomarker and its characteristics. You look for early biomarkers and look at studies that cite changes in lipid patterns or changes in quantity. These are excellent indices, but from there to being able to be used as a biomarker, a lot of work is involved in the development of biomarkers. Please discuss a little more the characteristics or properties of a biomarker so that the reader of your paper can come to the same conclusions you have made. I suggest a couple of reviews where you address the topic of biomarkers.
- Califf RM. Biomarker definitions and their applications. Exp Biol Med (Maywood). 2018 Feb;243(3):213-221. doi: 10.1177/1535370217750088. PMID: 29405771; PMCID: PMC5813875.
- Strimbu K, Tavel JA. What are biomarkers? Curr Opin HIV AIDS. 2010 Nov;5(6):463-6. doi: 10.1097/COH.0b013e32833ed177. PMID: 20978388; PMCID: PMC3078627.
We thank the Reviewer for this useful suggestion. We added an accurate definition of biomarker with its characteristics, properties and its classification to emphasize the importance of their use, from a preclinical to a clinical context (Page 3 line 114-142). In this way, we hope we have made our purpose clearer to the readers.

Reviewer 2 Report (New Reviewer)
In this review article, the authors summarized some novel lipidomic biomarkers of cerebrovascular diseases, which is an important clinical need. However, the presentation need extensive improvement. Also, it is essential to strengthen the background and discuss parts to enhance the scientific value.
1. Abstract: The lines 14-18 are about hemodynamic mechanisms of stroke and cerebrovascular diseases. It is not the main focus of this paper but essential as a background knowledge. I suggest remove it from the abstract and provide more details in the introduction.
2. Introduction: I would like to suggest simplifying the background of CNS and directly move to cerebrovascular disease. Three points need to be clarified. Firstly, there is an ethnic difference in the location of atherosclerosis in cerebral arteries, which is a major etiology of stroke. The intracranial atherosclerosis is commoner in Asian while plaques in extracranial carotid arteries are more frequently observed in Western countries (Refer: 10.1177/1747493016685716). Secondly, anatomic difference is an important consideration in evaluating cerebral circulation. On macrovascular level, the intact circle of Willis appear in only 50% of general population (Refer: 10.1109/ACCESS.2020.3007737). On microvascular level, parallel circulation especially the leptomeningeal collateral flow contribute significantly in sustaining cerebral perfusion in stroke (Refer: 10.1177/0271678X18805209). Finally, the biochemical and metabolic factors promote the development of intracranial atherosclerosis including in-stent restenosis, where the acceleration of transendothelial low-density lipoprotein (LDL) induced by abnormally low focal wall shear stress plays a key role (Refer: 10.3389/fneur.2022.1067566). However, although the risk of lipid disorder on cerebrovascular disease is well known, the specific lipid profile, e.g., the lipid molecules and their weight in the process, is still unclear, which limited the quantitative evaluation for clinical application.
3. This review is focused on the results in biochemical tests. It is well know that lipid component in atherosclerotic plaques is a direct indicator of stroke risk, and recent advances in imaging techniques including high-resolution MRI and CT enable the observation of lipid plaques (Refer: 10.1016/j.jacc.2013.06.015). The multidimensional morphological biomarkers including lipid features are more directly relevant to multiple clinical risks. This can be added in discussion at least.
4. The “methods”, i.e., literature search strategy, should follow the introduction part. I suggest to add a flow chart for literature screening.
5. As aforementioned, the discussion need to be enriched.
6. The paper need extensive revision in language and format, for example, the format of upper/lower letters in National Institute of Neurological disorders and strokes in lines 32-33.
The language is basically understandable but need further improvement. There are many minor errors in grammar and format.
Author Response
In this review article, the authors summarized some novel lipidomic biomarkers of cerebrovascular diseases, which is an important clinical need. However, the presentation need extensive improvement. Also, it is essential to strengthen the background and discuss parts to enhance the scientific value.
We thank the Reviewer for his/her valuable observations and suggestions. We agree with his/her suggestion to strengthen the background and the discussion in order to improve our work.
- Abstract: The lines 14-18 are about hemodynamic mechanisms of stroke and cerebrovascular diseases. It is not the main focus of this paper but essential as a background knowledge. I suggest remove it from the abstract and provide more details in the introduction.
As rightly suggested, we removed the lines 14-18 about hemodynamic mechanisms of stroke from the "Abstract" and better detailed this topic in line 36-41 page 1 (new version uploaded) in the first part of the “Introduction”.
- Introduction: I would like to suggest simplifying the background of CNS and directly move to cerebrovascular disease. Three points need to be clarified. Firstly, there is an ethnic difference in the location of atherosclerosis in cerebral arteries, which is a major etiology of stroke. The intracranial atherosclerosis is commoner in Asian while plaques in extracranial carotid arteries are more frequently observed in Western countries (Refer: 10.1177/1747493016685716). Secondly, anatomic difference is an important consideration in evaluating cerebral circulation. On macrovascular level, the intact circle of Willis appear in only 50% of general population (Refer: 10.1109/ACCESS.2020.3007737). On microvascular level, parallel circulation especially the leptomeningeal collateral flow contribute significantly in sustaining cerebral perfusion in stroke (Refer: 10.1177/0271678X18805209). Finally, the biochemical and metabolic factors promote the development of intracranial atherosclerosis including in-stent restenosis, where the acceleration of transendothelial low-density lipoprotein (LDL) induced by abnormally low focal wall shear stress plays a key role (Refer: 10.3389/fneur.2022.1067566). However, although the risk of lipid disorder on cerebrovascular disease is well known, the specific lipid profile, e.g., the lipid molecules and their weight in the process, is still unclear, which limited the quantitative evaluation for clinical application.
We are greateful to the reviewer for such suggestion so we shortened the background of CNS removing unnecessary information from line 40 to line 57 on page 2. We also agree with the three higlighted points so we added what suggested regarding ethnic differences in location of atherosclerosis (line 54-57 pag 2, ref 10.1177/1747493016685716 ), anatomic differences in evaluation cerebral circulation (line 59-65 pag 2, ref 10.1109/ACCESS.2020.3007737) and, finally, biochemical and metabolic factors promoting the development of intracranial atherosclerosis (line 100-105 pag 3, 10.3389/fneur.2022.1067566).
- This review is focused on the results in biochemical tests. It is well know that lipid component in atherosclerotic plaques is a direct indicator of stroke risk, and recent advances in imaging techniques including high-resolution MRI and CT enable the observation of lipid plaques (Refer: 10.1016/j.jacc.2013.06.015). The multidimensional morphological biomarkers including lipid features are more directly relevant to multiple clinical risks. This can be added in discussion at least.
We agree with the reviewer and we think this is a good point. We added two references about atherosclerotic plaques as direct indicator of stroke risk (10.1016/j.jacc.2013.06.015 as suggested and 10.1016/j.neurad.2018.05.003 as a more recent review on the same topic) expanding this point in line 183-188 page 4 of the new version.
- The “methods”, i.e., literature search strategy, should follow the introduction part. I suggest to add a flow chart for literature screening.
We thank the reviewer for this valuable advice. We moved the “Methods” after the introduction part. We also added a flow chart to illustrate our literature screening process.
- As aforementioned, the discussion need to be enriched.
As rightly suggested, we enriched the discussion in sereval points: line 54-57 pag 2, ref. 10.1177/1747493016685716, line 59-65 pag 2, ref. 10.1109/ACCESS.2020.3007737, line 100-105 pag 3, 10.3389/fneur.2022.1067566, line 114-142 ref. 10.1177/1535370217750088 and doi:10.1097/COH.0b013e32833ed177.
- The paper need extensive revision in language and format, for example, the format of upper/lower letters in National Institute of Neurological disorders and strokes in lines 32-33.
Comments on the Quality of English Language
The language is basically understandable but need further improvement. There are many minor errors in grammar and format.
We thank the reviewer for this advice. Our paper has been submitted to an extensive revision in language and format.

Round 2
Reviewer 2 Report (New Reviewer)
Thanks for the update. The majority of my earlier comments have been well addressed. However, I noted there are some minor errors in reference, language, and format. Please carefully compare the order of reference in response and manuscript, and improve the details in proofreading.
The English has been improved while further polishing is needed.
Author Response
Open Review
( ) I would not like to sign my review report
(x) I would like to sign my review report
Quality of English Language
( ) I am not qualified to assess the quality of English in this paper
( ) English very difficult to understand/incomprehensible
( ) Extensive editing of English language required
( ) Moderate editing of English language required
(x) Minor editing of English language required
( ) English language fine. No issues detected
|
Is the work a significant contribution to the field? |
|
|
Is the work well organized and comprehensively described? |
|
|
Is the work scientifically sound and not misleading? |
|
|
Are there appropriate and adequate references to related and previous work? |
|
|
Is the English used correct and readable? |
Comments and Suggestions for Authors
Thanks for the update. The majority of my earlier comments have been well addressed. However, I noted there are some minor errors in reference, language, and format. Please carefully compare the order of reference in response and manuscript, and improve the details in proofreading.
We thank the reviewer for pointing out the oversights in the references due to the reworking of the manuscript. The reviewer’s attention to detail was very helpful in arranging the text. Therefore, we checked and organized the exact order of references. We also made some changes to improve the language and format, rewieving the English text with the help of expert bilingual neurologists. We are confident that these changes have made the manuscript clearer to the reader.

This manuscript is a resubmission of an earlier submission. The following is a list of the peer review reports and author responses from that submission.
Round 1
Reviewer 1 Report
Dear Authors.
Thank you very much for writing this review that attempts to raise information about lipidomics and its use for biomarkers of cerebrovascular diseases.
The background provided in this review shows us different lipids associated with cerebrovascular diseases, this review, and the associated pathologies are interesting and well-represented in the development of the manuscript. However, a biomarker is more than a concentration of a biomolecule different from that observed in a control individual. It has certain characteristics, and certain conditions, which are not mentioned, nor developed by the authors. Furthermore, it is confuse what type of biomarker they wish to describe, whether it is a diagnostic, predictive, risk, or monitoring biomarker.
The concept of associating lipids as biomarkers is very innovative, and can really be a success in biomedicine, but it is not well developed or well achieved in this manuscript.
I urge the authors to rethink the idea, develop it in a better way, and receive another manuscript written in a better way.
Reviewer 2 Report
To write a good review, one needs to have a clear aim, consequently a review should start with an overview of what reviews exist already on the topic and how your review compares to these, e.g. is it an update, or an analysis of the existing knowledge from a different point of view. Next is a description on the literature search strategy, how, where and what period, did you search for the information, i.e. databases, keywords, etc. It is also important to describe what criteria were used to exclude publications from the review, e.g. not relevant for the aim of your thesis, or poor quality. Reviews on the same topic, or closely related topics should be mentioned in the introduction, with a clear explanation what your review adds to these already published reviews. A review should be more than a compilation of the results reported in literature (i.e. just copy and paste), it should in fact be a critical assessment of the present knowledge with some clear conclusions what all these results mean, and what are the perspectives and directions for future research and potential applications.
So what is the real idea of a review: the reader should afterwards be able to make decisions about what directions to go, what has already been shown to be not promising and what looks promising. What is needed is to make a critical assessment of all the data you have collected. It means that all the available data are integrated and those which are not fulfilling certain minimal quality requirements are pointed out. A review should be a source of inspiration and not just be a collection of facts, it should contribute to the discussion in the field, to be critical and thus result in improving the field.
Reviewer 3 Report
In this review article, the authors discuss lipidomics and cerebrovascular diseases. They proposed mass spectrometry technique as a tool for biomarkers discovery. The authors indicate that it is of great importance to identify new circulating biomarkers able to distinguish and monitor common and rare cerebrovascular diseases that lead to stoke events.
Overall, the review is too superficial and need extensive review of the body literature on lipidomics and cerebrovascular diseases that lead to stroke events.
1. Line 3, Title: please correct the error “biomarkery”
2. Abstract is unclear and need to be revised to reflect the body of the proposed review.
3. Introduction should include diet (ie, high fat diet) and metabolic syndrome as a risk factors for cerebrovascular diseases (CVD). Arachidonic acid metabolites should also be added and discussed.
4. Authors should discuss the etiology, epidemiology and pathophysiology of cerebrovascular diseases.
5. Sentence in line #44-47 should be revised to include disease cause as well supported with references.
6. Line # 68-69: This sentence needs a reference “So far, 30% of stroke events remains to be established, since their undetermined origin.”
7. Sentence in line # 69-70 (ref. #7) is unclear.
8. Figure 1 is too vague and needs clear description of the process.
9. Figure 2 needs description and moved up (in Section 2, line #125). Also, changes should be supported by references in each case.
10. Line #149-154: Authors should provide details of the fifteen plasma lipids reported here. They should also be careful about speculating on the translational aspect of findings from this single study cited here.
11. Line # 156: Section 2.2 should be split into 2 sub-sections.
12. Section 3.3 should be moved with CSVD.
13. Authors should provide a critical review of the body literature on lipidomics and cerebrovascular diseases. If indeed mass spectrometry is the tool for discovering these biomarkers, then the rational of selecting this technique among many others need to be provided and widely discussed in their review.
14. Please summarize the evidence supporting lipidomics and cerebrovascular diseases in a table/figure. This information will be valuable for researchers in the field.
15. Please summarize techniques to study lipdomics in animals and humans in a table. This information will be valuable for researchers in the field.
16. Please summarize key findings on lipid metabolites that cross the BBB in brain CVD in a table.
Reviewer 4 Report
The review is well compiled and informative.